# Places of safety? Fear and violence in acute mental health facilities: A large qualitative study of staff and service user perspectives

**Gabrielle Jenkin** [1]*, **Stewart Quigg**[1], **Hannah Paap**[1], **Emily Cooney**[1], **Debbie Peterson**[2], **Susanna Every-Palmer** [1]

**1** Department of Psychological Medicine, University of Otago, Wellington, New Zealand, **2** Department of Public Health, University of Otago, Wellington, New Zealand

* gabrielle.jenkin@otago.ac.nz

**Data Availability Statement:** The minimal dataset is provided in the manuscript and the supporting files (the codebook). The data codebook

## Abstract

### Aim

To understand violence on acute mental health units according to staff and service user perspectives and experiences.

### Background

The collateral damage of violence in acute inpatient mental health settings is wide-ranging, impacting on the health and wellbeing of staff and service users, and detrimental to public perceptions of people who are mentally unwell. Despite international research on the topic, few studies have examined psychiatric unit violence from both staff and service user perspectives.

### Methods

We conducted in-depth interviews with 85 people (42 staff, 43 service users) in four adult acute mental health inpatient units in New Zealand. We undertook a thematic analysis of perspectives on the contributing factors and consequences of violence on the unit.

### Results

Both staff and service users indicated violence was a frequent problem in acute inpatient units. Four themes regarding the causes of violence emerged: individual service user factors, the built environment, organisational factors, and the overall social milieu of the unit. Staff often highlighted complexities of the system as causal factors. These included the difficulties of managing diverse service user illnesses within an inadequate and unsafe built environment whilst having to contend with staffing issues and idiosyncrasies relating to rule enforcement. In contrast, service users talked of their needs for care and autonomy not being met in an atmosphere of paternalism, boredom due to restrictions and lack of meaningful activities, enforced medication, and physical confinement as precipitants to violence. Two broader themes also emerged, both relating to empathy. Both staff and service users exhibited 'othering' (characterised by a profound lack of empathy) in relation to acutely

(containing deidentified and aggregated data) is available as a Supporting Information file. In line with the ethics approval from HDEC ref (17/CEN/94), the full interview transcripts cannot be shared publicly. This is because of their highly sensitive nature and the personal accounts which make them potentially identifiable. For data inquiries please contact the University of Otago Human Ethics Committee, Academic Committees Office, 1st floor, Scott/Shand House, 90 St David's Street, Dunedin, New Zealand, or through the following webpage: https://www.otago.ac.nz/council/committees/committees/HumanEthicsCommittees.html.

**Funding:** This research was funded by a Marsden Fast Start from the Royal Society of New Zealand awarded to Gabrielle Jenkin (contract UOO1623). https://www.royalsociety.org.nz/what-we-do/funds-and-opportunities/marsden The funders had no role in study design, data collection and analysis, decision to publish, or preparation of the manuscript.

**Competing interests:** The authors have declared that no competing interests exist.

unwell individuals. Explanations for violent behaviour on the unit differed between groups, with service users being more likely to attribute unwanted behaviour to contextual factors and staff more likely to 'blame' mental illness. The consequences of violence included stress, physical injury, and a culture of fear and stigma.

## Conclusion

Violence in acute inpatient mental health units in New Zealand is a significant, complex, and unresolved problem negatively impacting the therapeutic mission of these settings. Further in-depth qualitative investigations are urgently required into what is experienced as violence by service users, their view of how violence occurs, the role of fear and power relations, and the contributions of the built and organisational environment to all forms of violence to all unit users. A core function of the acute mental health unit is to offer a therapeutic environment for individuals at their most vulnerable. For this to happen, the unit must be a rewarding place to work, and a safe place to be.

## Introduction

The relationship of serious mental illness to violence is a vexed one. While public perception often holds that an inexorable link exists between the two, [1, 2] most people with serious mental illness are *not* violent [3]. In fact, people with mental illness are more likely to be victims of violence than perpetrators, although this attracts less attention in the literature [4–6] and the media [7, 8].

Many studies investigating the relationship between serious mental illness and violence have identified a positive association between the two variables [3, 9, 10]. However, there is a wide variation in rates depending on definitions and measures of violence, study design, individual, and contextual factors and setting. The acute mental health inpatient unit is one such setting that has consistently been established as a high-risk environment [4].

### Violence in the acute inpatient setting

The acute inpatient psychiatric setting has an elevated risk of violence perpetrated by service users compared with community settings [4]. Nonetheless, it is clear most people admitted to acute psychiatric units do not behave violently. A meta-analysis of the rates of violence on acute psychiatric units in high-income countries found 17% (95% CI 14–20%) of service users threatened or committed a violent act (physical violence towards others) during their admission while 83% of those admitted were not violent [11]. Reported rates vary widely, with studies with higher rate of people with alcohol use disorders, male gender, schizophrenia, and compulsory treatment reporting higher rates of inpatient violence by service users. Young adulthood, single status, and a past history of violent and self-destructive behaviours have also been associated with higher rates of inpatient violence by service users [12, 13]. Research has also considered the role of smoking as a factor contributing to violence on inpatient units [14].

In a national audit of violence on inpatient units in the United Kingdom, one-third (36%) of psychiatric inpatients had experienced, and almost half (46%) had witnessed violence during their current inpatient stay [15]. In another retrospective United States study, one-third of the service user participants had been victims of violence as inpatients and about two-thirds had witnessed traumatic events [16]. Thirteen percent reported being physically assaulted (defined as being hit, punched, slapped, kicked, strangled, or burned) by a staff member and

26% reported being assaulted by another patient. Eighteen percent reported witnessing a staff member physically assault another service user and 39% had witnessed a service user assaulting someone. Eight percent reported at least one sexual assault in a psychiatric setting with a staff member being the perpetrator in 3% [16].

Female patients may be particularly vulnerable, with an Australian study finding that across nine different inpatient units, 85% of female inpatients reported feeling unsafe during hospitalisation, 67% reported experiencing sexual or other forms of harassment, and 45% experienced a sexual assault during an inpatient admission [17].

Overall, experiences of violence in inpatient units appear common across jurisdictions, with these experiences taking the form of service user-to-staff violence, service user-to-service user violence, and staff-to-service user violence [11, 18, 19]. There is some evidence to suggest that violence is more prevalent in the psychiatric setting compared to other specialities [20, 21]. A review of the nursing literature on workforce violence found that staff in psychiatric settings were the healthcare professionals most likely to have experienced physical violence at work, with incidence rates at 55%, followed by staff in the emergency and geriatric settings at 50% and 46% respectively [21].

## Staff and service user perspectives on violence

Historically, much of the research on violence in this setting has been quantitative; however, a number of qualitative studies have examined staff or service user perceptions and experiences of violence in the acute inpatient setting.

Due to long exposure and the nature of their role, mental health nurses may be more at risk of violence than other members of the multidisciplinary team [22]. Most mental health nurses report having been assaulted on acute inpatient units at some point during their career [22, 23]. In a multinational study, 75% of mental health nurses had experienced an assault at least once [24] and similar rates have been reported by psychiatrists [25]. In another study, 62% of mental health clinical staff reported being assaulted at least once in their careers and 28% reported an assault within the last 6 months, with high rates of associated PTSD symptoms [23]. Earlier research indicates that staff tend to believe people's mental illness has caused violent behaviour [26, 27].

The smaller pool of studies examining the service user perspective suggests high rates of traumatic and potentially harmful experiences for those admitted to psychiatric units, sometimes termed *sanctuary trauma* and *sanctuary harm* respectively–adverse experiences occurring inside an environment that should be a place of safety [16]. The power imbalance that exists within mental health units is often identified as a contributor to institutional violence from the perspective of people admitted to these units [28]. Service users highlight that staff violence may be veiled within the paradigm of coercive 'treatment', including compulsory treatment, seclusion, and restraint. For example, in a UK study, service users described restraints as violent acts perpetrated by staff in response to non-compliance, sometimes occurring with undue force ("*I wasn't restrained, I was attacked*" [29]). Furthermore, some service users identified certain staff as instrumental in provoking situations that made violence more likely [29]. Additionally, service users see environmental factors and poor quality of care as influential in their experience of violence [26, 30].

Some studies have conjointly considered staff and service user perspectives of violence on inpatient units. However, these have often either been restricted to forensic settings or considered specific incidents [31]. The problem is seen as a complex one, with institutional control via entrenched and hierarchical structures considered important in contributing to an environment in which violence occurs [32, 33].

### Impact of violence

Consequences of violence in the acute mental health setting are wide-ranging, and include physical, psychological, professional, social, and societal impacts. These include the immediate physical effects, and the subsequent psychological effects on the victim and others present, including the perpetrator. For staff, violence or the threat of violence can evoke fear and hyper-vigilance, affecting their practice [34]. Staff may implement coercive or restrictive measures in response to violence, resulting in restraint, enforced medication, or seclusion. These measures can further traumatise service users and perpetuate the cycle of violence [35]. Poor staff wellbeing, difficulty recruiting and retaining staff, higher service costs, and reduced standards of care are all collateral damage impacting both staff and service users [22, 23, 36, 37].

Although there are a couple of exceptions, very little has been published on service user experiences of violence on the unit, and its impacts on service users [28, 29]. Such research highlights how service users experience staff control, restraint, and enforced medication as violence and provocation. From the service user point of view, violence in this context can be and is seen by them as self-defence and a cry for help.

Because violence in the acute mental health setting remains an unresolved problem with serious adverse impacts, and due to the lack of qualitative research in this area, we decided to examine both staff and service user perspectives on the causes and consequences of violence in this setting, drawing on interview data from New Zealand case studies. A more thorough understanding of violence on psychiatric inpatient units would assist in its mitigation by informing changes to the physical and social infrastructure. Currently, the lack of a nuanced understanding of violence on inpatient units serves the status quo, and not until the various determinants of violence are explicated, can they be addressed. Thus, it is the endeavour of this paper to uncover some of the underappreciated complexities of violence and the interactions that coalesce to cause it.

## Methods

As part of a larger study on the architectural design and social milieu of the adult acute mental health unit in New Zealand, we sought to examine staff and service user perspectives and experiences of safety on the unit. Initially, we were not specifically focused on the phenomenon of violence on the unit. However, it soon became apparent that this was one of the most pressing issues affecting participants.

### Ethics, consultation, and locality approvals

Prior to funding, we undertook consultation with Otago Ngāi Tahu Research Consultation Committee as per University of Otago requirements for research involving Māori (the indigenous people in New Zealand). Ethics approval was received from the Central Health and Disability Ethics Committee in 2017 (17/CEN/94). We obtained locality consent from the four participating DHBs. The study protocol is available in the Australian and New Zealand Clinical Trials Registry: http://www.ANZCTR.org.au/ACTRN12617001469303.aspx. The names of the units and their precise locations have been de-identified as per our research protocol and assurance of confidentiality.

### Data collection

The collection of the data for this project required multiple site visits by the lead author, GJ, to conduct interviews on the four inpatient units and to collect other sources of data (not reported here) during 2017–2019. Data was collected from four different publicly funded

inpatient units from across New Zealand. We maximised case study diversity [38] using building age, condition, and location as criteria. The first four units prioritised for inclusion in our study agreed to participate.

## Recruitment

**Staff.** We invited staff from the full range of occupations on the acute mental health unit to participate in our study. These included nurses, nursing care assistants, social workers, occupational therapists, psychiatrists, pharmacists, clinical team leaders, and cultural advisors.

**Mental health service users.** As per ethics approval, we were provided with a list of service users on the unit who were competent to consent, well enough for the interview, and potentially interested in participating. The lead author then invited those on the list to participate in the research, interviewing those who provided written consent.

## Interview schedule

As part of a semi-structured interview schedule covering a wide range of topics, we asked staff and service user participants if they felt safe on the unit, and for service users, if they felt their belongings and personal effects were safe. While this provided some data on staff and service user perspectives on the issue of violence on the unit, compelling data also emerged spontaneously in response to other questions pertaining to rules of the unit; what happens in an emergency, and (in particular) rules concerning smoking and leave. The full interview schedules are provided in S1 File (with relevant questions in bold).

## Interviews

All interviews were conducted by the lead author, GJ, a social scientist, and experienced qualitative researcher and interviewer. GJ has not worked in or received care in any of the mental health units studied. GJ had not met any of the research participants prior to the interviews. The units studied were spread across New Zealand. GJ spent at least a week at each of the sites collecting data.

Most interviews took place face to face on the unit, with a few interviews conducted by phone (five staff and five service users) for those participants who had wished to participate, but only became available or well enough to do so after the interviewer had left the city. No staff were present for the service user interviews, and the interviews took place in a quiet room, usually located at the entrance to the unit.

For practical and budget reasons, the number of interviews conducted on each unit was capped at a maximum of ten each for staff and service users for each of the four units. However, considerable interest in participating resulted in a few further interviews. Interviews lasting around 30 minutes for service users and 30–90 minutes for staff were audio recorded and translated verbatim. Excerpts for the transcripts focused on safety and violence on the unit were extracted by GJ and HP and identified by consensus.

Across the four units, 85 interviews were conducted; 43 with service users and 42 with staff (see Table 1 for participant attributes).

Of the 43 service users, 34.9% were Māori (and the highest proportion of Māori were in Unit D) and 16.7% of staff (n = 7) were Māori. The proportion of Māori staff mirrors population ethnicity demographics, with 16.5% of New Zealanders identifying as Māori [39]. The higher percentage of Māori service users compared with population demographic likely reflects the finding that Māori are overrepresented as recipients of psychiatric inpatient care in New Zealand, for example Māori 2.9 times more likely to be subject to an indefinite inpatient treatment order than non-Māori [40].

**Table 1. Sample and case study characteristics.**

| *Participants* | Unit A | Unit B | Unit C | Unit D | Total |
|---|---|---|---|---|---|
| Service users | 10 | 11 | 12 | 10 | 43 |
| (%* female) | (50%) | (46%) | (58%) | (50%) | 51% |
| Staff | 9 | 13 | 11 | 9 | 42 |
| (%* female) | (44%) | (54%) | (64%) | (100%) | 64% |
| Total | 19 | 24 | 23 | 19 | 85 |
| *Ward characteristics* | | | | | |
| Beds | 22 | 64 | 21 | 32 | |
| Ward location | Hospital grounds | Own campus | Hospital grounds | Hospital grounds | |
| Geographic location | Major city in North Island | Major city in South Island | Small town in North Island | Major city in North Island | |

*% rounded

Staff participants comprised nurses or nurse managers (n = 20), social workers (n = 5), psychiatrists (n = 4), cultural or consumer advisors (n = 4), occupational therapists (n = 4), pharmacists (n = 2), and a doctor, cleaner, and a music therapist.

We did not collect data on psychiatric diagnoses. This was deliberate–we did not want service user participants to feel as if their experiences were being interpreted through an illness lens. Although national data on diagnoses at admission to an acute mental health facility are not publicly available, one study in Auckland New Zealand found the common discharge diagnoses to be mood disorders, including bipolar disorder (manic, depressive, and mixed episodes) depression, and psychotic disorders, such as schizophrenia and schizoaffective disorder [41]. New Zealand has a low number of acute inpatient beds per capita compared with other OECD countries [42], with high demand and occupancy reported [43]. Service users need to present with high acuity to access a psychiatric bed in New Zealand; therefore, the service users participating in this study likely represented a group with severe mental illness, dominated by mood and psychotic disorders.

## Analysis

Extracts from the transcripts were coded into themes separately by GJ, HP, and SQ using the iterative process described by Braun and Clarke [44]. The entire multi-disciplinary team of researchers, comprising a social scientist, a social anthropologist, two psychiatry doctors (collectively with 25 years' experience working on mental health units), a psychologist, and a mental health service user academic, then met over several workshop sessions to refine, clarify, and agree on the key themes. For rigour, the coding schedule with the diverse range of example quotes are provided as a (S2 File).

## Results

### Perception of violence

Violence was reported as a **substantial problem** for both staff and service users on all four acute mental health units. Participants reported assaults occurred on a regular basis, with some resulting in injury:

> *One of the nurses was walking past one of the bathrooms. And this patient was watching and as she walked past . . . opened the door out, so she smacked into the door. (staff A1)*

Staff and service users held different perspectives as to who was responsible for violence and what it looked like. Furthermore, on all four units service users were reluctant to inform on others, while assaults involving staff were recorded more frequently than those between service users (although it was unclear why):

*We get more records of interactions between nurses and patients that are problematic, because they record them in the book. But if incidents between patients are to be recorded, one of the patients will have to report it to a nurse, and it thought to be serious enough to record in the incident recording system. So, staff... might not actually report it. (staff B13)*

**Causes of violence.** Four broad themes (see Table 2) were identified by both staff and service users as responsible for violence: individual service user factors, the built environment, organisational factors, and the social milieu of the unit. Within these, sub-themes were identified, such as psychosis under individual factors. Service user and staff perspectives were consistent in some areas and divergent in others.

*I. Individual service user factors*. Themes around **service user illness,** and particularly **psychosis,** were consistently associated with safety concerns and violence. Staff identified psychiatric symptoms, especially paranoid delusions as a risk factor for violence:

*Because of their delusional beliefs, or their paranoia, or they believe that that staff member has done something to them, and they're going to get them back. (staff A3)*

Some staff explained how some services users were loud and disinhibited (lacking in self-control) when they were unwell, which could antagonise people:

*Sometimes a patient's illness will cause them to behave in a way that causes other people to want to hit them. (staff A3)*

Service users agreed that psychosis contributed to issues around safety. They identified other clients as being 'scary' due both to their own paranoia and the other person's unpredictable behaviour:

*It's very dangerous and they [people with psychosis] are also . . . easily angered because of the pent-up frustration from hearing the voices. (service user D3)*

Staff identified features of **service user histories** they considered increased the risk of violence, including past criminal offending, histories of antisocial behaviour, and/or previous violence:

*Forensics [the forensic mental health service] wouldn't take them, you know. So, they're really violent people. (staff A1)*

At times, some service users' behaviours that resulted in staff injury were perceived as **instrumental**—a way of getting what they wanted:

*I remember one guy hit [male staff member] with a chair. Knocked him out too. And he did it simply because he wanted to [go to] jail. (staff A1)*

**Table 2. Summary of the causes of violence.**

| Subthemes | Perspective | Description of theme |
|---|---|---|
| **Causes of violence** | | |
| ***Individual service user factors*** | | |
| Illness/psychosis<br>Disinhibition<br>Service user histories<br>Instrumental (goal achievement) | Staff | Service user illness, particularly psychotic symptoms and disinhibition, were considered risk factors for violence. Service user histories with apparent criminal and forensic backgrounds were also thought to be contributory. Some service users were seen as using violence to obtain a desired outcome. |
| Illness/psychosis<br>Self-perception | Service users | Service users also identified psychotic features as contributing to safety concerns. Occasionally, service users bragged about their violence, justifying it as a way of asserting dominance and control (being 'top dog') |
| ***Built environment*** | | |
| Confined spaces<br>Blind spots<br>Proximity between certain areas<br>Access to alarms<br>Insufficient exits<br>Lack of visibility<br>Temperature and ventilation | Staff | General unit design and layout issues were seen as instrumental in the occurrence of violence. Specific features included insufficient space, blind spots, and a lack of exits. |
| Confined spaces<br>Nurses' station design | Service users | Similar to staff views, confined spaces were seen as an aggravating factor. The physical and symbolic separation between service users and staff due to the fishbowl design of the nurses' station caused fear and safety concerns, and also contributed to violence. |
| ***Organisational factors*** | | |
| Smoking & rules<br>Staffing (adequacy, skills and experience, gender mix) | Staff | Lack of access to smoking was seen as a cause of violence. Inconsistent enforcement of smoking rules exacerbated matters. Wider issues, such as staffing, also impacted the safety of the unit in terms of cover, skills, experience, and gender mix. |
| Smoking & rules<br>Staffing | Service users | Service users held a similar sentiment to staff around smoking and its management, but identified inconsistent application of rules in general as being linked to violence. They too observed staffing issues. |
| ***Social milieu*** | | |
| Complexities (diverse service user illness/needs) | Staff | Different service user presentations and needs were related to violence e.g. managing elderly females with young and unpredictable males in the same space. |
| Locked unit<br>Paternalistic atmosphere<br>Boredom<br>Restraint, seclusion, and medication | Service users | Confinement through the locking of doors and the reported paternalistic culture maintained by staff were seen to lead to violence. This was either through acting out, or in response to boredom, which at times was met with restraint, seclusion, and enforced medication from staff (which itself was experienced as staff violence). |
| **Meta-themes** | | |
| Interpretations of behaviour | | |
| 'Othering' | | |

*And he said, "I'll do what the hell I want to get what I want, and if that means I assault you I will do, you fucking bitch." (staff A3)*

One service user participant relished their history of violence, indicating that this might be embedded in their identity or **self-perception**:

*They know I'm top dog everywhere I go, and they respect that–there was one day, I got so mad, I punched all the security guards there. (service users C7)*

*II. Built environment.* Staff highlighted that unit design and layout issues led to situations which compromised safety. This was a strong theme in three of the units studied, but was less prominent in the unit that had been built recently, which was seen as an environment that catered better to service users' needs. In the other units, staff commonly reported that the buildings were poorly designed and institutional, leading to **confined spaces** for service users:

*The closed ward should really have the bigger spaces to separate people. (staff D9)*

Further, design faults produced '**blind spots**' where there was poor visibility on the unit. This led to staff feeling fearful of being assaulted out of line of sight of their colleagues:

*I can remember this particular time, her talking about assaulting me, where we were in the high needs' unit . . . I suddenly thought that nobody in the nursing station can see me. This is quite scary, and my stomach was doing something, and she was looking at me with this look she had. (staff C11)*

Issues of **proximity between certain areas** on the units, **access to alarms,** and **insufficient exits** were also seen by staff to compromise care and present safety concerns. One staff member attests to 'dragging' service users some distance whilst under restraint to place them in seclusion. In that unit there was considerable distance between the open unit and the seclusion wing, with the path between the two via a long semi-public corridor.

Several staff described the risk of being 'trapped' or 'barricaded' in certain areas that lacked exits; these were usually in nurses' stations located in, or next to, seclusion areas of the unit:

*Doesn't have an exit door from that [area], which is problematic at times. You can be quite vulnerable if someone's really aggressive, and you have to retreat to the nurses' station, and you've got nowhere you can go. You've just got to hope the door holds. (staff B3)*

A staff member recalls this dynamic lead to a significant assault for one service user:

*One person has been locked in HCA [the high care area], a patient, and they got very badly assaulted by [another] patient. (staff B5)*

Staff were concerned for service user safety due to poor unit design. Staff cited issues with responding to 'fights' where they had difficulty locating events due to a **lack of visibility**:

*And I've been working on shifts where there's been fights in the corridor, and we're all looking . . . you can hear the noise, and there's like this full-on brawl between two males, . . . but we didn't know where it was. (staff B2)*

Poor **temperature control** was seen by staff as a 'trigger' for violent behaviour. Several staff identified inadequate air conditioning as contributing to violence; this was particularly evident in the confined spaces of the seclusion wing in one unit, and lack of ventilation was highlighted in another unit. Subsequent adverse outcomes related to safety were evidenced for both service users and staff:

*So, there's no air circulating. Too condensed, and it's hot. . . and his behaviour started escalating within half an hour of being in that area.. . . and then one of the staff was assaulted. (staff A2)*

Service users highlighted issues with **confined spaces** in this regard:

*Let us have fucking air. And it's usually because we're hyperventilating. We just need to go outside and have cool air, to just calm down. (service user B1)*

The **nurses' station design** was also an important factor. The three older units had traditional 'fish-bowl' nurses' station designs, where the nursing staff were based in a partially transparent 'bubble' looking out on the communal areas of the unit. These nurses' stations functioned as a place of congregation for service users requesting basic necessities from staff, such as medication, a hairdryer, or a phone charger.

The fishbowl design is a physical and symbolic barrier separating staff and service users. Service users reported that sometimes in times of crisis, staff 'locked' themselves safely inside the nurses' station while service users in most units could not lock their own bedrooms doors without staff assistance were not given the same opportunity. This made many service users frightened and exposed:

*And that's not nice to be told that they're fearful for their safety while you're standing on the other side of the glass and you're actually quite terrified yourself. (service user A6)*

Furthermore, there was evidence from service users' that this dynamic contributed to acts of violence:

*It's all shatterproof now because people used to throw chairs through it. (service user A2)*

*III. Organisational factors.* All four of the units studied had attempted to ban **smoking**; however, only one unit had successfully achieved this. When the other units had previously tried to become smoke-free, staff reported that violence increased:

*I know that it was tried a few years ago, before my time, and the assaults went berserk. So even though it's a non-smoking ward, people do smoke. (staff B9)*

**Rules around smoking** were one of the most commonly identified causes of violence by both staff and service users from all of the units studied:

*Because if these guys can't smoke, well they take it out on us, basically. (staff A1)*

Service users also recalled incidents of violence that had resulted from denying service users cigarettes:

*One guy went off his nut the other day before yesterday and started screaming at one of the elderly women because he wanted to go out for a smoke, and they wouldn't open the door until a certain time. So, he lost it, and threatened her. (service user A3)*

Several service users said **inconsistent application of rules** in general was a major stressor that contributed to violence. They said that the rule enforcement depended on the "mood of the staff." They also believed "who's rostered on at the time" influences "what the rules are."

Some service users felt this was unfair and "confusing because you don't really know what's going on." Some staff agreed rules were implemented inconsistently, which caused conflict:

*So, some staff are reinforcing [the smoke free rules] to the letter and other staff letting it slide. . . . For example, you'd have a client that is allowed to smoke all day long and then a change-over of shift the nurse that comes on [says], "You know this is a no-smoking area" . . . before you know it there's a bit of a tiff between . . . the staff member and the client and then the staff member feels like that the staff member on the earlier shift had set them up for, for all of this. (staff C10)*

Inconsistency of rule enforcement was often related to staffing issues. These issues in turn were viewed to have impacts on the safety of the unit. **Adequate staffing** was a common contributing factor as to whether staff felt safe at work:

*When there is enough staff, you feel safe. (staff D1)*

Units being understaffed did not go unnoticed by service users:

*All the staff are lovely, wonderful, but there needs to be more of them. Because some of them are doing two shifts. (service user A4)*

*They're completely unqualified and understaffed.. . . [They need] more staffing, because they can't always. . .you know, people are waiting. You'll wait there, and wait there, and wait there, until somebody looks at you. (service user A9)*

**Staff skills** and **experience** also determined whether staff felt safe or not. The more experience a staff member had working on acute mental health units, the safer they felt. Some believed that this made them more equipped to deal with incidents like violence when they occurred:

*I feel reasonably safe. But then again, I've been in these sort of situations for a long time. I know there's a lot of staff who've not long been nursing, or have only got four to five years' experience, where they still feel scared and upset when an incident occurs. (staff A3)*

The **staff gender mix** also contributed to how safe people felt. Often, there were significantly more female staff members on shift than males. This, according to staff, puts all staff under considerable pressure:

*We're a 90 per cent workforce female, and 10 per cent male. So, you can imagine the pressure that goes on those female staff, and the pressure that goes on the 10 per cent of males. (staff B3)*

Some male staff argued they were more likely to be victims of violence than female staff:

*You also get the ones that are more likely to take you on, just because they don't want to hit the small female, but they've got no qualms hitting the big guy. . . . I have been attacked a lot of times. (staff A7)*

*IV. Social milieu.* Both staff and service users noted the **complexities** of the unit and the difficult and potentially dangerous dynamic that results. On their part, staff talked of differences

in background, experience, and training whilst acknowledging the eclecticism in presentations of the service users whom they cared for. This included differences in age, gender, ethnicity, and understanding of mental illnesses and its manifestations. Staff described:

*We're managing aggressive, aggro males that have used drugs, and have no belief that they're unwell, with your grandmother. (staff B3)*

Service users held the same sentiment:

*[I feel unsafe because] the mixing of the males and females and the drug addicts. (service user A2)*

A key issue of concern for many service users was that of physical confinement. Many staff believed **locking the unit** was necessary to either stop vulnerable service users from leaving or to help 'paranoid' service users "feel more comfortable." For some service users this was analogous to being 'caged' and impacted on their wellbeing with the perception of being imprisoned:

*This is supposed to be a health facility. . . [but] I feel like I'm in prison. (service use A6)*

Multiple service users agreed and commented on the effects of being infantilised in this **paternalistic atmosphere**:

*Sometimes, when they just treat us like kids, and we get angry they'll just be like, "Oh, have some medicine. That'll make you better." So, "No, it won't." (service user B3)*

*Because this is not reality. This is like a forced. . . you're a child. You're reduced to being three years old. You're getting your nappies changed, sort of shit. It's just real bad. You have to ask for pull-ups, or you have to ask for your period pads and stuff like. . . for an independent adult, it's really hard to regress back to a child state, where you're banned from going outside. (service user B1)*

Often confined to the unit, service users recalled their dislike of what they experienced as an impoverished environment. Many complained of **boredom** and a lack of meaningful activities, which frustrated them and contributed to their risk of violence:

*There's nothing to do, you get bored, you get agitated. My partner has ADHD and he needs to be doing stuff. And the only programs they have here are for three-year olds. (service user A9)*

Occasionally service uses talked of resorting to violence as a means of acting out against these conditions. Other service users recalled how **restraint, seclusion, and medication** were used as a means of control over them. They described treatment being forced on them or others, and experienced this as staff violence:

*One person had a fit, a hissy fit, and made a ruckus. But he was pinned down and injected with a drug to calm him down. (service user B6)*

*Meta-themes*. The above quotes also highlight two meta themes identified from the data, both relating to empathy. These are not directly linked to the causes of violence but occur throughout participant accounts. Staff often framed their **interpretations of behaviours** on

the unit in terms of illness-related symptoms and with reference to past experience. In contrast, service users' explanations were more contextualised with their, and their fellow inpatients', current lived experience on the unit Service users appeared to have appreciation for nuanced interpersonal dynamics between those on the unit, including power differentials existing both within service users as a group, and between staff and service users.

The second meta theme relates to the process of '**othering**,' referring to the tendency to assign negative characteristics to a certain group, individual, or community based on perceived differences between the assigner and the othered. The assigner attributes events or behaviour that they dislike to the othered's membership of that group [45–47]. This leads to (or amplifies) marginalisation and inequality. At times, both staff and service users separated the unwell 'others' from everyone else. Most frequently this was using generalised terms, such as 'they' or 'them,' but also more explicitly using terms like 'drug addicts' and 'crazy' [sic] in pejorative and discriminatory ways. Additionally, both staff and service users ascribed non-human characteristics to people who were unwell. One service user recalled:

> *If you are having a psychotic breakdown like he was, it's just quite blood rage. You could see in his face and he was just puffed up like a blimmin' silverback gorilla, basically that's what he looked like. (service user A2)*

Whilst a staff member recounted an interaction with a service user under the apparent influence of substances:

> *When I have had a 90 kg guy throw one of our staff members who is 204 kg against the wall during drug-induced psychosis. That is superhuman strength that we're dealing with. How am I supposed to restrain that? (staff D2)*

**Consequences of violence.** Both staff and service users talked about the toll violence took on them, physically, mentally, and professionally (see Table 3).

Staff felt resigned to the risk of violence, seeing it as "part of their job," indicating a degree of **normalisation**:

> *When you come into work in an HDU [high dependency unit], you accept those things. You know some things are going to happen. At some point during your career you're going to be on ACC [receiving accident compensation] because you've been punched in the face. (staff D2)*

Staff often felt **unsupported by management** and policy makers, who made decisions in relation to violence and safety without visiting the units:

**Table 3. Summary of the consequences of violence.**

| Consequences of violence | |
|---|---|
| Staff | Normalisation—violence is "part of the job" |
| | Unsupported by management |
| | Self-perpetuating problem (retention and recruitment difficulties, comprised training) |
| | Injuries |
| | Time off work |
| | Mental health issues |
| Service users | Fear |
| | Impeded recovery |
| | Stigma |

*I just hate when people are writing policies when they've never even stood foot on an HDU [high dependency unit] ward. (staff D2)*

Some also felt under too much 'responsibility' and 'pressure,' making them consider other careers, especially considering safety concerns:

*Because [of] the staff safety issues–I think a lot of people have got sick of that and just gone, "You know what, it's not worth it," and just, "I'm going to work at a day-care." (staff D2)*

Staff alluded to the above issues as presenting a **self-perpetuating problem**, creating retention and recruitment difficulties, and also compromising time available for staff training:

*We're really low on staff and we've such high staff turnover, a lot of our extra learning opportunities are getting taken away because we don't have enough people on the ward to take us off the ward to do for the education. (staff D2)*

Staff members reported high rates of **injuries** requiring **time off work** on Accident Compensation Corporation (ACC; employee and employer levied system in New Zealand). Some of the injuries were described as severe:

*We have three people with brain injuries off [work] at the moment. (staff D2)*

Stress and distress were common consequences of feeling unsafe. Some staff experienced their own **mental health issues**, which they attributed to work stress:

*Like, I was actually meant to be on night shift this week, but I had a panic attack and couldn't continue. (staff B2)*

Staff used various coping strategies, some of them unhealthy:

*I light up and have a cigarette while I'm walking to work. Anxious. (staff A1)*

Service users noticed that staff were under high stress, and some service users expressed considerable empathy:

*Look, the staff are really good, you know, fantastic, understaffed. . . . There needs to be an admin person . . . the desk full time, because you have to take your razor there and your cell phone chargers, everything you have to get here. So, you've got people all day long asking for these things and it's taking away precious time for the nurses that need it. (service user A4)*

*She [another service user] gets quite nasty to the staff. Quite sad. I said to the staff the other day, . . . "We love you [name], don't worry." Because she gave [them] a hard time. (service user A4)*

Unsurprisingly, violence on the units also caused **fear** amongst some service users, making them feel vulnerable:

*Sometimes I feel as though they're going crazy. I feel threatened. (service user B5)*

Being scared also **impeded recovery**. For one service user in particular, being afraid made her illness worse:

*If I go in a psychosis and I'm scared, paranoid and scared, then it makes me even sicker.*

She continued:

*It can be the wrong place to be. I've got post-traumatic stress, so yeah, a scary situation like that can be really bad. (service user A8)*

Some service users felt **stigmatised** by the way staff perceived them:

*It's stigma, yeah, yeah. It feels like they're working with prisoners. It almost feels like we're criminals, even though, some of us can function, and others can't. Different degrees, obviously, on the spectrum but at the end of the day, it just feels like they don't want to be near you. You know, everything's blocked off, everything's, yeah, it's just, it's really, really is over the top. (service user A2)*

## Discussion

Historically, violence on acute inpatient mental health units has been well documented and our study provides further evidence this remains an unresolved issue adversely impacting on all the people who inhabit those spaces. We found four key themes as causes of violence on the unit: individual service user factors, the built environment, organisational factors, and the overall social milieu of the unit. We found more nuanced data in the sub-themes.

Staff believe that they are faced with a near impossible proposition of managing the acutely unwell in an inadequate environment, whilst trying to keep themselves and vulnerable service users safe with insufficient resources. Staff identified many catalysts for violence, including acute psychosis, disinhibition, smoking, and the complexity of service user needs. Additionally, staff believed that safety was compromised by specific issues within the built environment, such as blind spots, lack of exits, confined spaces, and heating and ventilation issues. Staffing issues relating to staffing adequacy, skill, experience, and gender mix were also seen as contributory to violence.

In contrast, service users described a sense of vulnerability and need for care, expressing frustration with inconsistent rules, especially around smoking [48], and boredom from lack of activities as amplifying their distress. They also identified staff separation, exemplified by the nursing station 'fishbowl' design, as creating further risks, leading to anger, and feeling ignored and abandoned. Service users expressed that feeling imprisoned, infringements on autonomy, other restrictions, and coercive treatment (including restraint, seclusion, and forced treatment) were contributors, and even provocations, to violence. Some service users experienced controls as forms of violence from staff. Legal sanctions to manage and control service users through restrictive models of care mean that staff violence to service users may be dismissed, being ignored as acceptable, necessary, and in everyone's best interests. For example, the staff members in this study who talked about 'dragging' service users some distance to the seclusion wing, viewed this in terms of preventing violence rather than committing violence. This coercive framework was highlighted by Hamrin et al. [49] who revealed how power struggles lead to violence on inpatient units, and Omérov et al.'s [32] finding that the majority of patients believed they were provoked prior to violent incidents, most often by staff.

Although a common narrative was that individual service user characteristics (illness or personality) were the root causes of violence, on critical examination, we think that this has been overstated. This draws upon stigmatising stereotypes and may be an attempt to make sense of violence on the mental health units. It also alleviates personal responsibility. As noted earlier, the link between violence and mental illness has been overstated and people with mental illnesses are much more likely to be victims of violence than perpetrators [3, 5]. We also note that while using individual factors to explain inpatient violence has been the dominant discourse in much of the quantitatively dominated research in this field, this work and view is outdated due to its serious lack of investigation into service user experiences of violence and the organisational, social, and built environment of mental health units.

Our comparison of service user and staff perspectives on violence also highlighted two meta-themes, both relating to empathy. Staff empathic explanations of behaviour they disliked tended to focus on mental illness symptoms. In contrast, service users' empathic explanations of behaviour they disliked (both from other service users and staff) were more nuanced, rich, and contextually detailed. In the taxonomy of validation, staff explanations focussed on biology and past learning, whereas service users' explanations tended to be normative and focussed on the here-and-now [50]. The second meta-theme relates to the tendency for both groups to 'other' individuals who have been violent e.g. by comparing them to animals, and otherwise lumping them within a group and assigning that group negative characteristics. This polarises opinion, stigmatises people, and blocks understanding and intervention, some of which service users talked about in relation to the consequences of violence.

The consequences for both staff and service users were significant and multi-dimensional, mirroring those found by previous research [22, 23, 31, 35–37, 51]. Our findings showed that staff suffered both physical and mental health consequences leading to more stringent use of seclusion and even job departure. Staff described knock on-effects with less skilled staff being employed to fill vacancies, leading to further cycles of vulnerability and violence. Service users, in turn, talked of feeling stigma from their experiences on the inpatient unit, and also the feeling of fear, which impeded their recovery. Violence on the inpatient unit, whether perpetuated by, or witnessed by service users, will affect their experiences of mental health services in the future, potentially cause PTSD, physical injury, and distrust of services and health professionals. Furthermore, service users may face criminal charges for their violent actions, which they might feel unfair in instances where they felt it was justifiable self-defence.

Results were broadly consistent across the four inpatient units with one notable exception. One of the facilities was a new build and had been constructed using co-design principles incorporating service user, community, cultural, and staff perspectives. From this unit, service users expressed fewer concerns about safety and staff focussed on organisational factors, particularly understaffing, as the key determinants of risk, rather than environmental factors. For the other units, there were considerably more data on how inadequate design and an impoverished environment negatively impacted the social milieu, creating feelings of boredom, frustration, and fear. This suggests that newer co-designed mental health units may reduce violence or at least some of the factors contributing to it, which deserves further qualitative and quantitative exploration.

## Strengths and limitations

A major strength of this study is its twin focus on both staff and service users' views on violence. Historically, most perspectives have come from nursing staff [52]. Our research was able to encapsulate the diversity of opinion for staff on the inpatient unit including specialties, such as occupational therapy and medicine. The inclusion of service users' views alongside

clinicians for general adult mental health units is rare [26, 53], and enabled us to derive novel results, including new understanding of causes of violence and perpetuating factors, such as staff shortages and compromised training. The inclusion of multiple sites and large numbers of participants expands the generalisability of these findings.

Another strength lay in the interviewing and data analysis methods. Often within a research groups, the interviews are undertaken by junior members of the team. In this study, all the interviews were conducted by a social scientist with over 25 years' experience in qualitative methods. This rendered high quality, in-depth transcripts for analysis. GJ's perceived position as relative 'outsider', neither a service user nor a health professional, but a respectful and skilled listener who allowed the participant to be the expert of their own experience, proved to be a powerful tool for eliciting rich and nuanced data about a sensitive subject. The data analyses underwent many iterations due to discussions between research team members who brought multiple perspectives including social anthropology, clinical psychology, inpatient psychiatry, and academic service user experience. The team comprised researchers of mixed ages, experiences, and gender, from health and non-health backgrounds, and included researchers who had provided and who had received inpatient care. This contributed to rich, in-depth, and interesting discussions that added to our understanding of the multiple perspectives and experiences reported in the data. This aided the iterative process of identifying the most salient themes and our choice of words used to describe these.

This study includes purely qualitative data and, as such, is subject to the limitations of this field. Notably, staff participants self-selected, and service user participants were screened by staff (as well enough to participate) after self-selecting. Both staff and service user at times gave answers that were contradictory. Some services users were also still quite unwell at interview and occasionally there were responses that were reflective of this.

A further limitation was that because interviews were primarily conducted in the inpatient unit and staff clearly have the power in this environment, service users may not have felt entirely free to speak candidly about their experience on the unit. Given recent service user-led research emerging in the area [28, 29], service user perspectives on violence on the inpatient unit are urgently needed. These should include their perspectives on what violence looks like, causes and contributing factors, the impact of various coercive practices, and restrictions on their choices, autonomy, wellbeing, and recovery. To capture these views requires some in-depth qualitative interviews with service users, preferably conducted after discharge, to reduce the power differential.

## Implications

Similar to other studies [28, 29], our service user participants described power imbalances on the unit as contributors to institutional violence. Environmental factors were identified as major determinants, which also aligns with other studies [26, 30]. Previous research has suggested approaching the issue of violence on mental health units via a virtue ethics paradigm through the creation of therapeutic environments [49]. This entails moving away from rule-based approaches and rather creating spaces fostering therapeutic relationships. Our findings support this rationale. These results also substantiate the importance of previously identified pragmatic issues, such as creating meaningful engagement between staff and service users, reduced coercive practice, and enhanced staff training [26].

Environmental and systemic issues require further attention [54, 55]. The outcomes from this study suggest that interventions at these levels may have the most benefit. Services users could be provided with a more fit-for-purpose built environment designed to promote recovery [56]. This should be informed by insights and needs of the building users, staff, service

users, and family. This will likely include re-design of current facilities to smaller, home style environments to foster positive engagement between staff and service users [57], open designs without visible nurses' stations that place nurses alongside service users, and designs to accommodate greater choice of therapies, meaningful activities, lines of sight and personal security for service users (a lock on their bedroom door and a safe place or locker to store personal items). Service user views on inpatient unit violence should be urgently sought to understand how much of service user violence arises from fear [29]. Additionally service user views on solutions to the problem of violence should also be sought, initial work in this area has been undertaken pointing to changes to the built environment, staff and service user relations and a system to address and monitor complaints [28].

## Conclusion

The experience of both staff and service users is that inpatient mental health units are volatile spaces punctuated by violence. Staff provided dramatic insight into the chronicity of safety issues, highlighting the toll it has taken on the workforce. This is juxtaposed with the acute needs of service users, including the scared and disempowered, those with psychotic symptoms, and individuals with criminal backgrounds, who are braggadocios about their use of violence. When all housed together in the combustible environment of an acute inpatient unit, our research has evidenced that the outcomes are often detrimental, with acts of violence committed by all parties against all parties. With the determined goal of inpatient units to providing safety and care for those in need, these findings have left us to question are these places of healing or of custody?

Violence on acute mental health inpatient units is a significant and complex issue adversely impacting on the therapeutic goal of these settings. Issues, such as inadequate resources, poorly designed environments, coercive frameworks, and lack of service user perspectives and input on the models of care and the built environment, are challenges we must embrace. There must be integrative and holistic solutions where management, staff service users, and their family collaborate. This may include reversing traditional roles where staff and service users have greater input in the allocation of funds where management spend more time at the 'ground level,' and staff have some induction training to spend time living on the unit as a service user to gain insight into the service user experience. A further imperative is the redesign of units so that they are fit for purpose.

## Supporting information

**S1 File. Interview schedules.**
(DOCX)

**S2 File. Coding schedule.**
(DOCX)

## Acknowledgments

We thank the four participating District Health Boards and the research participants for their engagement and participation in this research.

## Author Contributions

**Conceptualization:** Gabrielle Jenkin.

**Data curation:** Gabrielle Jenkin, Hannah Paap.

**Formal analysis:** Gabrielle Jenkin, Stewart Quigg, Hannah Paap, Emily Cooney, Debbie Peterson, Susanna Every-Palmer.

**Funding acquisition:** Gabrielle Jenkin, Debbie Peterson.

**Investigation:** Gabrielle Jenkin, Stewart Quigg, Hannah Paap, Emily Cooney, Debbie Peterson, Susanna Every-Palmer.

**Methodology:** Gabrielle Jenkin.

**Project administration:** Gabrielle Jenkin, Susanna Every-Palmer.

**Resources:** Gabrielle Jenkin.

**Supervision:** Gabrielle Jenkin, Hannah Paap.

**Validation:** Gabrielle Jenkin.

**Writing – original draft:** Gabrielle Jenkin, Stewart Quigg, Hannah Paap, Emily Cooney, Debbie Peterson, Susanna Every-Palmer.

**Writing – review & editing:** Gabrielle Jenkin, Stewart Quigg, Hannah Paap, Emily Cooney, Debbie Peterson, Susanna Every-Palmer.

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
