## [Decision Letter · Decision Letter 0]

7 Jan 2022

PONE-D-21-16154Places of safety? Fear and violence in acute mental health facilities: a large qualitative study of staff and service user perspectivesPLOS ONE

Dear Dr. Jenkin,

Thank you for submitting your manuscript to PLOS ONE. After careful consideration, we feel that it has merit but does not fully meet PLOS ONE’s publication criteria as it currently stands. Therefore, we invite you to submit a revised version of the manuscript that addresses the points raised during the review process. The two reviewers have made several suggestions to improve the paper.

We look forward to receiving your revised manuscript.

Kind regards,

Nancy Beam, PhD

Staff Editor

PLOS ONE

Journal Requirements:

“This research was funded by a Marsden Fast Start from the Royal Society of New Zealand (UOO1623).”

“This research was funded by a Marsden Fast Start awarded to Gabrielle Jenkin from the Royal Society of New Zealand (contract UOO1623).

https://www.royalsociety.org.nz/what-we-do/funds-and-opportunities/marsden

Reviewers' comments:

Reviewer's Responses to Questions

**Comments to the Author**

1. Is the manuscript technically sound, and do the data support the conclusions?

Reviewer #1: Yes

Reviewer #2: Yes

2. Has the statistical analysis been performed appropriately and rigorously? 

Reviewer #1: N/A

Reviewer #2: Yes

3. Have the authors made all data underlying the findings in their manuscript fully available?

Reviewer #1: Yes

Reviewer #2: Yes

4. Is the manuscript presented in an intelligible fashion and written in standard English?

Reviewer #1: Yes

Reviewer #2: Yes

5. Review Comments to the Author

Reviewer #1: Jenkin, et al. completed qualitative interviews with staff and service users in acute mental health facilities. The authors found that both staff and service users are concerned about violence within the wards. Interestingly, both staff and service users identified issues within the facilities (such as understaffing) and specific symptoms of service users (particularly psychosis) as concerning. However, staff were more likely to blame mental illness while service users attributed problems to contextual factors.

Overall, this paper has a number of strengths. This is a clearly written paper with a large sample size. Perceptions of violence in acute psychiatric wards of both staff and service users is an important and relevant topic for many facilities. In particular, this paper is strengthened by its interviews of service users, highlighting their lived experience. For example, it is an important point from service users that staff may not be recording violence committed by staff towards service users. There are clear themes and excellent quotes to support themes throughout the manuscript.

However, this paper does have some weaknesses. It would be helpful to include rates of violence in other inpatient units to compare to rates of violence in psychiatric units in the introduction. The paragraph at the end of page 5 regarding service user perceptions should be edited to be more clear and could be expanded, since the experience of the service user is an important part of this paper. I would like to see a bit more on the discrepancy between Māori service users and staff (34.9% of service users and only 16.7% of staff) and the charged language staff used when describing Māori service users (briefly noted on page 19).

This paper is an important contribution to the literature on violence in acute mental health facilities, particularly due to its sample size and interviews of service users.

Reviewer #2: Thank you for the opportunity to review your manuscript. The paper is well written, benefits from a clear description of the qualitative analysis, and has some important implications for the design of acute inpatient facilities. I think the paper would be suitable for publication with some minor revisions. Below are my comments and suggestions for each section of the paper:

Introduction – you start by suggesting that the relationship between severe mental ill-health and violence is not supported by evidence and then continue to reference several papers and reviews that suggest a clear link between the two (particularly in terms of service user victimisation) and state that “Many studies investigating the relationship between serious mental illness and violence have identified a positive association between the two variables”. You then end this section with “with these experiences taking the form of service user-to-staff violence, service user-to-service user violence, and staff-to-service user violence”. This narrative didn’t flow well for me, and I wondered whether it should be changed a little – you are suggesting that there is a link between severe mental ill-health and violence and that this is indeed supported by the evidence. Why do you suggest that it isn’t supported by evidence at the beginning?

I think more could be done to justify the rationale for this qualitative study. It’s a really interesting piece of work but the rationale needs to be clearer and it might also benefit from a more in depth discussion of the other qualitative research on service users/providers perspectives (a lot of time is spent referencing percentages and quant work at the beginning!).

Methods – this section is good and it’s nice to see the authors describe who has conducted the analysis and the method involved (something other papers I’ve reviewed recently have failed to do!).

Some suggestions to further improve include providing more detail in relation to:

• The four acute mental health wards participants were recruited from – can you provide further details about these wards please?

• The interview topic guide – it might be useful to provide the full topic guide as an appendix highlighting the questions that were relevant for the current study?

• The relationship between GJ and the acute mental health wards/service providers/service users – it would be good to know whether GJ was already known by service providers/service users and in what capacity.

• How many interviews were conducted face to face and how many by phone. Why were some conducted by phone?

• The setting in which the interviews took place – you mention this as a key limitation in your discussion and this left me wondering whether service providers present during the interviews with service users? Which rooms did the interviews take place in? Who was present?

• Table 1 – is it possible to provide more information about service providers and service users, please? It would be useful to detail professional roles, psychiatric diagnoses, ages, etc. and split the table by acute ward (i.e., 1, 2, 3, 4). Were there any differences in demographics between wards? You may not be able to provide this information due to ethics but if you can I think this would be really interesting.

• Data saturation – was data saturation reached? I know you were constrained by the number of interviews you were able to undertake and that you conducted a few additional interviews. Was this because data saturation hadn’t been reached yet?

Results – I really enjoyed reading the results and thought these were presented well. I like the tables of themes and sub-themes and meta-themes discussed. The only real thing that was missing for me was a discussion of whether there were any differences in perspectives from the four different wards? Also, was there anyone that had any contradictory views to those presented about the violence on wards?

Discussion – The discussion is good, and I particularly like the implications section. Some suggestions which could help improve the discussion include:

• Implications section could be expanded to critically analyse whether suggested re-designs might be practical. Presumably they were designed that way for a reason?

• Discussion of how these findings might be relevant for other settings. This study was conducted in New Zealand. Are the findings likely to be relevant to other contexts?

• Discussion of how the causes of violence in the current study compare to those suggested in previous research.

General – there are some inconsistencies in punctuation before referencing quotes. Please could you correct these to be consistent throughout?

6. PLOS authors have the option to publish the peer review history of their article (what does this mean?). If published, this will include your full peer review and any attached files.

Reviewer #1: No

Reviewer #2: No

---

## [Author Response · Author response to Decision Letter 0]

23 Feb 2022

ONE-D-21-16154: Places of safety? Fear and violence in acute mental health facilities: a large qualitative study of staff and service user perspectives

PLOS ONE

10 February 2022

Dear Dr Nancy Beam, Staff Editor,

Thank you for providing us with the reviews and reviewer comments for our submission. The reviewers clearly had interest and expertise in the subject and we greatly appreciated them having read the article carefully and providing such constructive comments. We have addressed all the points raised and feel this has improved the quality of our manuscript.

We provide our comments to the reviewers in the letter below and the revised paper with track changes.

Kind regards,

Gabrielle Jenkin 

Please include the following paragraph as our amended funding statement: 

This research was funded by a Marsden Fast Start from the Royal Society of New Zealand (UOO1623). The funders had no role in study design, data collection and analysis, decision to publish, or preparation of the manuscript.”

4. Please review your reference list to ensure that it is complete and correct. Any changes to the reference list should be mentioned in the rebuttal letter that accompanies your revised manuscript. 

 

Author response to Reviewer #1:

Reviewer #1: Jenkin, et al. completed qualitative interviews with staff and service users in acute mental health facilities. The authors found that both staff and service users are concerned about violence within the wards. Interestingly, both staff and service users identified issues within the facilities (such as understaffing) and specific symptoms of service users (particularly psychosis) as concerning. However, staff were more likely to blame mental illness while service users attributed problems to contextual factors.

Overall, this paper has a number of strengths. This is a clearly written paper with a large sample size. Perceptions of violence in acute psychiatric wards of both staff and service users is an important and relevant topic for many facilities. In particular, this paper is strengthened by its interviews of service users, highlighting their lived experience. For example, it is an important point from service users that staff may not be recording violence committed by staff towards service users. There are clear themes and excellent quotes to support themes throughout the manuscript.

However, this paper does have some weaknesses. It would be helpful to include rates of violence in other inpatient units to compare to rates of violence in psychiatric units in the introduction. 

We agree that it would be useful to include rates of violence in mental health wards and in other inpatient wards; however, unfortunately there no data is available on rates of violence in inpatient wards in NZ. We also note that none of the other papers we had initially referenced offered comparisons of rates of violence across specialties/wards. Despite this, we have added to the introduction the following text to address this point, with two additional references: 

There is some evidence to suggest that violence is more prevalent in the psychiatric setting compared to other specialities. [20, 21] A review of the nursing literature on workforce violence found staff in psychiatric settings were those most likely to have experienced physical violence at work, with incidence rates at 55%, followed by staff in the emergency and geriatric settings at 50% and 46% respectively. [21]

The paragraph at the end of page 5 regarding service user perceptions should be edited to be more clear and could be expanded, since the experience of the service user is an important part of this paper. 

We agree this is an important aspect of the paper. We have clarified this paragraph. It now reads as follows:

The smaller pool of studies examining the service user perspective suggests high rates of traumatic and potentially harmful experiences for those admitted to psychiatric units, sometimes termed sanctuary trauma and sanctuary harm respectively – adverse experiences occurring inside an environment that should be a place of safety. [16] The power imbalance that exists within mental health units is often identified as a contributor to institutional violence from the perspective of people admitted to these units [28]. Service users highlight that staff violence may be veiled within the paradigm of coercive 'treatment’, including compulsory treatment, seclusion and restraint. For example, in a UK study, service users described restraints as violent acts perpetrated by staff in response to non-compliance, sometimes occurring with undue force (“I wasn’t restrained, I was attacked” [29]). Furthermore, some service users identified certain staff as instrumental in provoking situations that made violence more likely. [29] Additionally, service users see environmental factors and poor quality of care as influential in their experience of violence. [26, 30] 

I would like to see a bit more on the discrepancy between Māori service users and staff (34.9% of service users and only 16.7% of staff) and the charged language staff used when describing Māori service users (briefly noted on page 19).

We have added a paragraph explaining that the proportion of Māori staff reflects underlying population demographics and the proportion of Māori service users reflects health service demographics. Māori, like many Indigenous people, are overrepresented as mental health service users (likely due to the effects of colonisation and socioeconomic inequities; although, an in-depth discussion of this is beyond the scope of this paper) and this is a phenomenon already well described.

It was actually another service user who made a racist generalisation about Māori service users, not a staff member. Racism was coded to a single comment rather than being a recurrent theme throughout our 84 participants, but it came under the broader theme of ‘othering’. On reflection, we think using this as an example may raise too many other questions, and we have replaced this with another example.

This paper is an important contribution to the literature on violence in acute mental health facilities, particularly due to its sample size and interviews of service users.

Thank you!

Author response to Reviewer #2:

Reviewer #2: Thank you for the opportunity to review your manuscript. The paper is well written, benefits from a clear description of the qualitative analysis, and has some important implications for the design of acute inpatient facilities. I think the paper would be suitable for publication with some minor revisions. Below are my comments and suggestions for each section of the paper:

Introduction – you start by suggesting that the relationship between severe mental ill-health and violence is not supported by evidence and then continue to reference several papers and reviews that suggest a clear link between the two (particularly in terms of service user victimisation) and state that “Many studies investigating the relationship between serious mental illness and violence have identified a positive association between the two variables”. You then end this section with “with these experiences taking the form of service user-to-staff violence, service user-to-service user violence, and staff-to-service user violence”. This narrative didn’t flow well for me, and I wondered whether it should be changed a little –you are suggesting that there is a link between severe mental ill-health and violence and that this is indeed supported by the evidence. Why do you suggest that it isn’t supported by evidence at the beginning?

The key point was that although there are increased rates of violence within acute inpatient settings, the majority of people with serious mental illness do not behave violently and the relationship is more complex than some of the literature suggests. Our multidisciplinary authorship team felt it was important that our paper did not add to the stigmatising view that most people with serious mental illness are dangerous. We have edited this to try and make it clearer.

Our first few paragraphs now read as follows:

The relationship of serious mental illness to violence is a vexed one. While public perception often holds that an inexorable link exists between the two, [1, 2] most people with serious mental illness are not violent. [3] In fact, people with mental illness are more likely to be victims of violence than perpetrators, although this attracts less attention in the literature [4-6] and the media. [7, 8] 

Many studies investigating the relationship between serious mental illness and violence have identified a positive association between the two variables. [3, 9, 10]. However, there is a wide variation in rates depending on definitions and measures of violence, study design, individual, and contextual factors and setting. The acute mental health inpatient unit is one such setting that has consistently been established as a high-risk environment. [4]

Violence in the acute inpatient setting

The acute inpatient psychiatric setting has an elevated risk of violence perpetrated by service users compared with community settings. [4] Nonetheless, it is clear most people admitted to acute psychiatric units do not behave violently. A meta-analysis of the rates of violence on acute psychiatric units in high-income countries found 17% (95% CI 14–20%) of service users threatened or committed a violent act (physical violence towards others) during their admission while 83% of those admitted were not violent. [11] Reported rates vary widely, with studies with higher rate of people with alcohol use disorders, male gender, schizophrenia, and compulsory treatment reporting higher rates of inpatient violence by service users. Young adulthood, single status, and a past history of violent and self-destructive behaviours have also been associated with higher rates of inpatient violence by service users. [12, 13] Research has also considered the role of smoking as a factor contributing to violence on inpatient units. [14]

I think more could be done to justify the rationale for this qualitative study. It’s a really interesting piece of work but the rationale needs to be clearer and it might also benefit from a more in depth discussion of the other qualitative research on service users/providers perspectives (a lot of time is spent referencing percentages and quant work at the beginning!).

We have added a paragraph at the end of the introduction as further justification for this research. This paragraph reads:

Because violence in the acute mental health setting remains an unresolved problem with serious adverse impacts, and due to the lack of qualitative research in this area, we decided to examine both staff and service user perspectives on the causes and consequences of violence in this setting, drawing on interview data from New Zealand case studies. A more thorough understanding of violence on psychiatric inpatient units would assist in its mitigation by informing changes to the physical and social infrastructure. Currently, the lack of a nuanced understanding of violence on inpatient units serves the status quo, and not until the various determinants of violence are explicated, can they be addressed. Thus, it is the endeavour of this paper to uncover some of the underappreciated complexities of violence and the interactions that coalesce to cause it. 

Methods – this section is good and it’s nice to see the authors describe who has conducted the analysis and the method involved (something other papers I’ve reviewed recently have failed to do!).

Some suggestions to further improve include providing more detail in relation to:

• The four acute mental health wards participants were recruited from – can you provide further details about these wards please?

We have extended the table with more rows to include some of the key characteristics of the four case studies (bed numbers and locations). We cannot provide more detail as the ward names and precise locations have been de-identified as per our research protocol and assurance of confidentiality. (We have now noted this in the methods)

• The interview topic guide – it might be useful to provide the full topic guide as an appendix highlighting the questions that were relevant for the current study?

We have provided the full interview schedules as a supplementary file, with relevant questions bolded as suggested. 

• The relationship between GJ and the acute mental health wards/service providers/service users – it would be good to know whether GJ was already known by service providers/service users and in what capacity.

We have added the following comment:

All interviews were conducted by the lead author, GJ, a social scientist, and experienced qualitative researcher and interviewer. GJ has not worked in or received care in any of the mental health units studied. GJ had not met any of the research participants prior to the interviews. The units studied were spread across New Zealand. GJ spent at least a week at each of the sites collecting data.

• How many interviews were conducted face to face and how many by phone. Why were some conducted by phone?

We have added the following comment:

Most interviews took place face to face on the unit, with a few interviews conducted by phone (five staff and five service users) for those participants who had wished to participate, but only became available or well enough to do so after the interviewer had left the city. No staff were present for the service user interviews, and the interviews took place in a quiet room, usually located at the entrance to the unit.

• The setting in which the interviews took place – you mention this as a key limitation in your discussion and this left me wondering whether service providers present during the interviews with service users? Which rooms did the interviews take place in? Who was present?

We have added the following comment: 

No staff were present in the service user interviews, and the interviews took place in a quiet room usually at the entrance to the ward.

• Table 1 – is it possible to provide more information about service providers and service users, please? It would be useful to detail professional roles, psychiatric diagnoses, ages, etc. and split the table by acute ward (i.e., 1, 2, 3, 4). 

We have provided the following additional text in the methods: 

Staff participants comprised nurses or nurse managers (n=20), social workers (n=5), psychiatrists (n=4), cultural or consumer advisors (n=4), occupational therapists (n=4), pharmacists (n=2), and a doctor, cleaner, and a music therapist.

We did not collect data on psychiatric diagnoses. This was deliberate – we did not want service user participants to feel as if their experiences were being interpreted through an illness lens. Although national data on diagnoses at admission to an acute mental health facility are not publicly available, one study in Auckland New Zealand found the common discharge diagnoses to be mood disorders, including bipolar disorder (manic, depressive, and mixed episodes), depression, and psychotic disorders, such as schizophrenia and schizoaffective disorder [38]. New Zealand has a low number of acute inpatient beds per capita compared with other OECD countries [39] with high demand and occupancy reported [40]. Service users need to present with high acuity to access a psychiatric bed in New Zealand; therefore, the service users participating in this study likely represented a group with severe mental illness, dominated by mood and psychotic disorders.

Were there any differences in demographics between wards? You may not be able to provide this information due to ethics but if you can I think this would be really interesting.

There were no notable demographic differences between wards and we provided data by ward. We have provided what we can in Table 1 as we had to ensure that the wards are not identifiable. 

• Data saturation – was data saturation reached? I know you were constrained by the number of interviews you were able to undertake and that you conducted a few additional interviews. Was this because data saturation hadn’t been reached yet?

Data saturation was reached. However, we conducted additional interviews because we had invited a wider group of participants than required. We received more interest in participating than we had anticipated.

Results – I really enjoyed reading the results and thought these were presented well. I like the tables of themes and sub-themes and meta-themes discussed. The only real thing that was missing for me was a discussion of whether there were any differences in perspectives from the four different wards? Also, was there anyone that had any contradictory views to those presented about the violence on wards?

Yes, there was one notable difference between units. We have added the following paragraph

One of the facilities was a new build and had been constructed using co-design principles incorporating service user, community, cultural and staff perspectives. Overall, this unit was perceived as being more therapeutic and sensitive to its users’ needs. From this unit, service users expressed fewer concerns about safety and staff focussed on organisational factors, particularly understaffing, as the key determinants of risk, rather than environmental factors. For the other units, there were considerably more data on how inadequate design and an impoverished environment negatively impacted the social milieu, creating feelings of boredom, frustration, and fear. This suggests that newer more purpose-built mental health units may lessen violence or at least some of the factors contributing to it, which deserves further qualitative and quantitative exploration.

 Discussion – The discussion is good, and I particularly like the implications section. Some suggestions which could help improve the discussion include:

• Implications section could be expanded to critically analyse whether suggested re-designs might be practical. Presumably they were designed that way for a reason?

This is an interesting point. Looking at the design of some of these facilities, it is really not clear why they were built the way they were! A critical analysis of this is beyond the scope of this paper but this is a question we are exploring further with architect colleagues in another piece of research.

• Discussion of how these findings might be relevant for other settings. This study was conducted in New Zealand. Are the findings likely to be relevant to other contexts?

We consider the findings are likely to be applicable to other countries with a similar model of publicly funded acute inpatient mental health units that aim to admit and stabilise acutely unwell people before they are discharged back into the community, with a median length of stay of around two weeks.

• Discussion of how the causes of violence in the current study compare to those suggested in previous research.

We have expanded the following section:

Similar to other studies [28, 29], our service user participants described power imbalances on the unit as being contributors to institutional violence. Environmental factors were identified as major determinants, which also aligns with previous research [26, 30]. Previous research has suggested approaching the issue of violence on mental health units via a virtue ethics paradigm through the creation of therapeutic environments. [49] This entails moving away from rule-based approaches and rather creating spaces fostering therapeutic relationships. Our findings support this rationale. These results also substantiate the importance of previously identified pragmatic issues, such as creating meaningful engagement between staff and service users, reduced coercive practice, and enhanced staff training. [26]

General – there are some inconsistencies in punctuation before referencing quotes. Please could you correct these to be consistent throughout?

We have gone through and addressed the inconsistences in punctuation before referencing quotes.

Changes to the reference list:

The following references have been added to the reference list:

• 20. Perkins M, Wood L, Soler T, Walker K, Morata L, Novotny A, et al. Inpatient Nurses' Perception of Workplace Violence Based on Specialty. J Nurs Adm. 2020;50(10):515-20.

• 21. Spector PE, Zhou ZE, XX C. Nurse exposure to physical and nonphysical violence, bullying, and sexual harassment: a quantitative review. Int J Nurs Stud. 2014;51(1):72-84.39. Stats NZ. Māori ethnic group 2020. Available from: https://www.stats.govt.nz/tools/2018-census-ethnic-group-summaries/m%C4%81ori.

• 40. Ministry of Health. Office of the Director of Mental Health and Addiction Services Annual Report 2018 and 2019. Wellington: Ministry of Health, 2021.

• 41. Wheeler A, Robinson E, Robinson G. Admissions to acute psychiatric inpatient services in Auckland, New Zealand: a demographic and diagnostic review. N Z MED J. 2005;118(1226).

• 42. Mulder RT, Bastiampillai T, Jorm A, Allison S. New Zealand's mental health crisis, He Ara Oranga and the future. N Z MED J. 2022;135(1548):89-95.

• 43. Lewis O. Mental health unit overcrowding a torture convention breach 2021, Jun 17. Available from: https://www.newsroom.co.nz/mental-health-unit-overcrowding-a-torture-convention-breach.

The following reference has been updated from:

41. Jenkin G, McIntosh J, Hoek J, Mala K, Paap H, Peterson D, et al. There’s no smoke without fire: smoking in smoke-free acute mental health wards. PLoS One. forthcoming; submitted 12 April.

to:

48. Jenkin G, McIntosh J, Hoek J, Mala K, Paap H, Peterson D, et al. There’s no smoke without fire: smoking in smoke-free acute mental health wards. PLoS One. 2021;16(11):e0259984.

---

## [Decision Letter · Decision Letter 1]

31 Mar 2022

Places of safety? Fear and violence in acute mental health facilities: a large qualitative study of staff and service user perspectives

PONE-D-21-16154R1

Dear Dr. Jenkin,

We’re pleased to inform you that your manuscript has been judged scientifically suitable for publication and will be formally accepted for publication once it meets all outstanding technical requirements.

Kind regards,

Michelle Torok, Ph.D.

Academic Editor

PLOS ONE

Additional Editor Comments (optional):

Reviewers' comments:

Reviewer's Responses to Questions

**Comments to the Author**

1. If the authors have adequately addressed your comments raised in a previous round of review and you feel that this manuscript is now acceptable for publication, you may indicate that here to bypass the “Comments to the Author” section, enter your conflict of interest statement in the “Confidential to Editor” section, and submit your "Accept" recommendation.

Reviewer #1: All comments have been addressed

Reviewer #2: All comments have been addressed

2. Is the manuscript technically sound, and do the data support the conclusions?

Reviewer #1: Yes

Reviewer #2: Yes

3. Has the statistical analysis been performed appropriately and rigorously? 

Reviewer #1: Yes

Reviewer #2: Yes

4. Have the authors made all data underlying the findings in their manuscript fully available?

Reviewer #1: No

Reviewer #2: Yes

5. Is the manuscript presented in an intelligible fashion and written in standard English?

Reviewer #1: Yes

Reviewer #2: Yes

6. Review Comments to the Author

Reviewer #1: I do not have further comments and am excited to recommend this manuscript for acceptance. Thank you for your revisions.

Reviewer #2: Thank you for your response to my comments. You have addressed all the points raised and I am happy to recommend your manuscript for publication.

7. PLOS authors have the option to publish the peer review history of their article (what does this mean?). If published, this will include your full peer review and any attached files.

Reviewer #1: No

Reviewer #2: No

---

## [Editor Report · Acceptance letter]

8 Apr 2022

PONE-D-21-16154R1 

Places of safety? Fear and violence in acute mental health facilities: a large qualitative study of staff and service user perspectives 

Dear Dr. Jenkin:

I'm pleased to inform you that your manuscript has been deemed suitable for publication in PLOS ONE. Congratulations! Your manuscript is now with our production department. 

Kind regards, 

on behalf of

Dr Michelle Torok 

Academic Editor

PLOS ONE